# Remote Patient Monitoring with Wearable Sensors Following Knee Arthroplasty

**DOI:** 10.3390/s21155143

**Published:** 2021-07-29

**Authors:** Scott M. Bolam, Bruno Batinica, Ted C. Yeung, Sebastian Weaver, Astrid Cantamessa, Teresa C. Vanderboor, Shasha Yeung, Jacob T. Munro, Justin W. Fernandez, Thor F. Besier, Andrew Paul Monk

**Affiliations:** 1Department of Orthopaedics, Auckland City Hospital, Auckland 1023, New Zealand; s.bolam@auckland.ac.nz (S.M.B.); t.vanderboor@gmail.com (T.C.V.); jacob.munro@auckland.ac.nz (J.T.M.); 2Department of Surgery, University of Auckland, Auckland 1023, New Zealand; bbat644@aucklanduni.ac.nz; 3Auckland Bioengineering Institute, University of Auckland, Auckland 1010, New Zealand; tyeu008@aucklanduni.ac.nz (T.C.Y.); s.d.weaver@outlook.com (S.W.); syeu670@aucklanduni.ac.nz (S.Y.); j.fernandez@auckland.ac.nz (J.W.F.); t.besier@auckland.ac.nz (T.F.B.); 4Laboratory of Biological and Bioinspired Materials, University of Liège, 4000 Liège, Belgium; astrid.cantamessa@student.uliege.be; 5Department of Engineering Science, University of Auckland, Auckland 1010, New Zealand

**Keywords:** knee arthroplasty, wearable sensor, inertial measurement unit (IMU), PROMs, remote monitoring, telemedicine

## Abstract

(Background) Inertial Measurement Units (IMUs) provide a low-cost, portable solution to obtain functional measures similar to those captured with three-dimensional gait analysis, including spatiotemporal gait characteristics. The primary aim of this study was to determine the feasibility of a remote patient monitoring (RPM) workflow using ankle-worn IMUs measuring impact load, limb impact load asymmetry and knee range of motion in combination with patient-reported outcome measures. (Methods) A pilot cohort of 14 patients undergoing primary knee arthroplasty for osteoarthritis was prospectively enrolled. RPM in the community was performed weekly from 2 up to 6 weeks post-operatively using wearable IMUs. The following data were collected using IMUs: mobility (Bone Stimulus and cumulative impact load), impact load asymmetry and maximum knee flexion angle. In addition, scores from the Oxford Knee Score (OKS), EuroQol Five-dimension (EQ-5D) with EuroQol visual analogue scale (EQ-VAS) and 6 Minute Walk Test were collected. (Results) On average, the Bone Stimulus and cumulative impact load improved 52% (*p* = 0.002) and 371% (*p* = 0.035), compared to Post-Op Week 2. The impact load asymmetry value trended (*p* = 0.372) towards equal impact loading between the operative and non-operative limb. The mean maximum flexion angle achieved was 99.25° at Post-Operative Week 6, but this was not significantly different from pre-operative measurements (*p* = 0.1563). There were significant improvements in the mean EQ-5D (0.20; *p* = 0.047) and OKS (10.86; *p* < 0.001) scores both by 6 weeks after surgery, compared to pre-operative scores. (Conclusions) This pilot study demonstrates the feasibility of a reliable and low-maintenance workflow system to remotely monitor post-operative progress in knee arthroplasty patients. Preliminary data indicate IMU outputs relating to mobility, impact load asymmetry and range of motion can be obtained using commercially available IMU sensors. Further studies are required to directly correlate the IMU sensor outputs with patient outcomes to establish clinical significance.

## 1. Introduction

Post-operative monitoring of knee arthroplasty patients is essential to highlight avoidable complications that might benefit from early intervention. Early identification of patients with potentially devastating problems of infection and stiffness allows for timely intervention. Traditional outpatient schedules of follow-up provide limited opportunity for review. Furthermore, around 20% of patients are not satisfied after surgery and continue to have postoperative pain and functional limitations, with associated implications for recovery [1,2,3].

This suggests a need for optimised rehabilitation strategies that focus on early identification of patients on a suboptimal course through assessment of early post-operative physical function [4]. Methods to assess early post-operative physical function can be divided into three categories: (1) patient-reported outcome measures (PROMs) subjective to patient’s perception of function, activity, and symptoms; (2) performance-based outcome measures (PBOMs), which objectively assess physical capacity; and (3) wearable motion tracking sensors measuring physical activity [4,5,6,7].

Wearable sensors, such as Inertial Measurement Units (IMUs), provide a low-cost, portable solution for obtaining functional measures similar to those captured with three-dimensional gait analysis, including spatiotemporal gait characteristics [8,9,10,11,12]. In addition, IMUs can provide estimates of impact loading [13], step counts [14], loading asymmetry [15], and cumulative load exposure (combining both step counts and magnitude of impact load) [16]. IMUs provide an ideal opportunity for remote patient monitoring (RPM) in the community.

There is a potential role for IMUs to be used to supplement existing post-operative function measures (i.e., PROMs and PBOMs) and provide additional diagnostic value through clinic- and home-based RPM [7]. IMUs have been applied extensively in younger orthopaedic patients following sports injury [17,18,19,20] and are becoming increasingly common in evaluation of knee arthroplasty procedures [7,21,22,23,24]. However, many of these previous studies reported functional performance based on a single stand-alone function score, as opposed to using a variety of gait parameters, or have not combined IMU measurements with existing validated PROMS and PBOMs.

The purpose of this study was to assess the role of combining IMUs with PROMs and PBOMs to provide a viable alternative to the current standard of post-operative care following knee arthroplasty. We present a system that combines various functional metrics during walking gait, obtained from ankle-worn IMUs.

## 2. Materials and Methods

### 2.1. Study Participants and Design

Local ethics committee approval was obtained from The Auckland Health Research Ethics Committee (reference No. 000072).

Fourteen patients (male/female = 6/8) with end-stage degenerative change of the knee awaiting unilateral total knee arthroplasty (TKA) or unicompartmental knee arthroplasty (UKA) surgery were prospectively enrolled and followed up pre-operatively to 12 weeks post-operatively. The mean age was 66.8 ± 7.0 years, and the mean BMI was 30.6 ± 5.9 kg/m^2^. The American Society of Anaesthesiologist (ASA) classification system [25] showed 3/14 were ASA Class 1 (i.e., healthy person), 7/14 were ASA Class 2 (i.e., mild systemic disease) and 4/14 were ASA Class 3 (i.e., severe systemic disease). Nine patients underwent TKA and five patients underwent UKA. No perioperative complications were reported over the 12-week post-operative study period. Patient demographics and operation details are shown in Table 1.

The study design is outlined in Figure 1, consisting of baseline preoperative assessment of PROMs followed by post-surgery IMU data collected in the community after outpatient rehabilitation clinic and PROMs questionnaires completed pre-operatively and Post-Op Weeks 2–6 and 12. Additionally, the Six-Minute Walk Test (6MWT) was performed pre-operatively and at Post-Op Week 6 and maximum knee flexion angle was determined during walking and sit-to-stand exercises at these time points. All patients received physiotherapy for early mobilization with full weight bearing from Post-Op Day 1. From Post-Op Week 2, patients attended an outpatient rehabilitation program led by physiotherapists. Most patients completed the rehabilitation program in 4 weeks; however, some patients left early or took longer due to individual recovery timelines.

### 2.2. Patient-Reported Outcome Measures

Patients completed an online version of the Oxford Knee Score (OKS) [26,27] and EuroQol Five-dimension (EQ-5D) with the EuroQol visual analogue scale (EQ-VAS) scores [28] pre-operatively, at each outpatient rehabilitation session (Week 2–6) and then a final questionnaire at Week 12. The OKS were used as both are reliable, easy to administer and have been previously validated in patients with osteoarthritis [27]. The EQ-5D is a generic instrument designed to measure self-reported health-related quality of life (HRQoL) that measures five dimensions: mobility, self-care, usual activities, pain/discomfort, and anxiety/depression [28]. The 5-digit result of the EQ-5D questionnaire represents 1 of 243 possible health states. Each health state can be converted into a single summary index score (EQ-5D index score), ranging from 0.00 (worst possible health state) to 1.00 (best possible health state), by applying HRQoL weights from a valuation set. The valuation set used in this study was the commonly used UK EQ-5D Index Tariff, which is based on a large sample from the UK population [29]. The questionnaires were collected on a tablet (Apple iPad 7th Gen, Cupertino, CA, USA).

### 2.3. Remote Patient Monitoring with Ankle-Worn Inertial Measurement Units

Two ankle-worn IMUs were used to measure the linear accelerations and angular velocities of each limb in the frontal, sagittal and transverse planes (Vicon Blue Trident sensor, Vicon, Oxford, UK, www.imeasureu.com/imu-sensor/, accessed 1 February 2021). Each sensor connects via Bluetooth to a smart phone and is synchronised and triggered to record data onto an on-board Secure Digital card at a rate of 1149 Hz. After attending the outpatient physiotherapy clinic, IMUs were attached to each ankle over the anteromedial aspect of the distal tibia (2 cm cranial to medial malleolus) using a Velcro strap (Figure 2A).

Each IMU was triggered to record data for each session in the community via the application, Capture.U (Vicon, Oxford, UK) for ~8–10 h duration (outpatient rehabilitation clinic finished at midday). The IMU sensor is waterproof with a 12-h battery life, so patients were instructed to perform all routine normal activities, including washing and showering, taking the device off only prior to sleeping. The data on the IMUs were recorded to the device and uploaded when they were returned at the following outpatient clinic to the IMU Step desktop application (IMeasureU.com, Vicon, and Oxford, UK, accessed on 1 February 2021). The data were then viewed and analysed on the Cloud Dashboard. For each session, total impact load (g) and Bone Stimulus were analysed for each limb and impact load asymmetry between the operative and non-operative limb was calculated.

Impact load was calculated by the formula:$$\mathrm{Impact}\;\mathrm{load}=\sum n_{1g}+2*n_{2g}+3*n_{3g}+\cdots +x*n_{xg}$$
where *n_xg_* is the number of steps for a given impact level and *x* is the impact level from 1 to 200 g.

The Bone Stimulus metric is a metric of the cumulative load which describes the relationship between mechanical load stimuli and skeletal tissue remodelling as a function of stimulus. It is based upon the mechanobiology of bone and the premise that bone tissue responds more to the magnitude of stress, rather than the number of loading cycles [30]. Bone stimulus was calculated by the formula:$$\mathrm{Bone}\;\mathrm{stimulus}=\;{\left[\overset{k}{\underset{j=1}{\sum }}n_{j}{\left(\sigma_{j}\right)}^{m}\right]}_{per\;day}^{\frac{1}{2m}}$$
where *n* is the number of cycles, σ is the peak tibial acceleration (mechanical stress) and m is the empirical constant to take in to account the bone response.

Impact load asymmetry was calculated by the difference in average impact load per step between the operative and non-operative limb then divided by the sum of the average impact load of both limbs. Impact load asymmetry was calculated by the formula:$$\mathrm{Impact}\;\mathrm{load}\;\mathrm{asymmetry}=\frac{\bar{IL_{\mathrm{operative}}}-\;\bar{IL_{\mathrm{non}-\mathrm{operative}}}}{\bar{IL_{\mathrm{operative}}}+\;\bar{IL_{\mathrm{non}-\mathrm{operative}}}}\;\times \;100$$
where $\bar{IL_{left}},\;\bar{IL_{right}}$ is the average impact load per step (operative and non-operative leg, respectively).

### 2.4. Pre-Op and Post-Op Week 6 6MWT and Maximum Knee Flexion Angle

In 7 out of 14 total patients, a 6MWT and maximum knee flexion angle during walking and sit-to-stand exercises were performed pre-operatively and 6 weeks post-operatively. The 6MWT is a PBOM that measures the maximal walking distance covered in 6 min, supervised by a research assistant. Subjects were instructed to walk as far as possible in 6 min in a safe manner in an undisturbed 50-m loop track. The test is reliable in patients with knee osteoarthritis and after TKA [31]. To determine maximum knee flexion angle during functional, dynamic movements, for example walking and sit-to-stand exercises, two proximal sensors above the knee joint were taped on to each thigh (over the distal anterior femur, 5 cm cranial to the knee joint line), in addition to the two distal ankle-worn sensors (Figure 2B).

A machine learning algorithm was developed using the Python package ‘tsfresh’ to semi-automate feature selection and extraction from IMU data, as previously described [32].

To train this surrogate model initially, synchronised optical motion capture and IMU data were collected from four patients walking at a self-selected speed over ground. In brief, a 10-camera optical motion capture (OMC) system (Vicon MX Cameras, Oxford Metrics Group, Oxford, UK) and three force-plates (Berec Force Plates, Bertec, OH, USA) were used to collect motion data (100 Hz) and ground reaction force data (2000 Hz), respectively. Optical markers were placed on body segments of participants in accordance with the ‘University of Western Australia’ (UWA) marker set [32]. A scaled Open Sim model was used to obtain sagittal plane knee kinematics, which were used to train the surrogate model [33]. The training process for this surrogate model has been previously described [34].

The IMU data were windowed one second backwards from the time point of the target value. This forms a collection of the windowed IMU data paired with its target value. For each window of IMU data, we performed ‘tsfresh’ feature extraction, where over 700 features were calculated per channel of IMU data. We then used the ‘tsfresh’ feature selector to determine the feature’s significance in predicting the target value and removed any that were non-significant. Then a random forest regressor was trained using the selected features. To further reduce the number of features, we then used the model-based feature importance to determine the rank of each feature when used in random forest regressor model, then selected only the top 200 features, as previously described [35]. The reduced feature set was then used to generate the patient-specific models to estimate knee flexion angle and predict maximum knee flexion angle [36].

### 2.5. Privacy

All data from the IMUs were de-identified and stored on a cloud server compliant with the New Zealand Health Information Privacy Code 2020. The patient cohort was followed up for three months post-surgery with all data points stored on a dashboard visible to only the patient and researchers, using password-protected login credentials. Study patients were assigned a random patient identification (ID) number that was used for all documentation and further study analysis. The data collected from the application were associated with each participant from the user ID entered in the application. No participant’s personal health information data were logged at any point on the application or the smartphone.

### 2.6. Statistical Analysis

All results are shown as mean ± standard deviation (SD) of the mean, and all statistical analyses were performed using either Wilcoxon matched-pairs signed rank test or matched-pairs one-way analysis of variance (ANOVA) test with Tukey’s post hoc analysis. Statistical analyses and graphing were performed using the GraphPad Prism 8 software (GraphPad Software, San Diego, CA, USA). A *p*-value of <0.05 was considered statistically significant.

## 3. Results

### 3.1. Patient-Reported Outcome Measures

On average, the mean OKS exceeded the pre-operative score (25.60 ± 8.83) at Post-Op Week 3 (30.46 ± 8.20; *p* = 0.036) and continued to improve over the recovery period. The OKS exceeded the preoperative baseline by 39% and 52% at Post-Op Week 6 (36.46 ± 6.91 and Week 12 (38.92 ± 7.50), respectively. For the EQ-5D, the index score improved from pre-operative baseline (0.63 ± 0.191) at Post-Op Week 6 and 12 by 32% (0.83 ± 0.152; *p* = 0.047) and 37% (0.86 ± 0.152; *p* = 0.006), respectively. The EQ-VAS showed a strong trend in improvement from the pre-operative score (75.36 ± 17.86) over the recovery period, but did not reach statistical difference by Post-Op Week 12 (87 ± 9.53, *p* = 0.252) (Figure 3).

### 3.2. Remote Patient Monitoring with Ankle-Worn Inertial Measurement Units

The Bone Stimulus and impact load significantly improved (*p* < 0.05) from Post-Op Week 2 by Week 4 and Week 6, respectively. By Post-Op Week 6 the mean Bone Stimulus and impact load had increased 52% (90.73 ± 28.19 to 138.21 ± 19.70; *p* = 0.002) and 371% (1424.77 ± 1637.87 to 5290.01 ± 2606.97; *p* = 0.035), compared to Post-Op Week 2. Post-operatively, the average impact load of the non-operative limb was greater than the operative limb resulting in a negative average impact load asymmetry value from Week 2 to Week 6. From Post-Op Week 2 to Week 6, the impact load asymmetry value increased from −17.55 ± 16.50 to −10.62 ± 15.88, but did not reach statistical significance (*p* = 0.308) (Figure 4).

In the majority of patients, both the PROMs and IMU outputs gradually trended higher in value over the post-operative period, as expected. This is exemplified in Patient 1, where there is a gradual, consistent improvement in the PROMs scores (i.e., OKS) and Impact load (Figure 5). However, in the example of Patient 2, despite the patient reporting improving OKS scores between Post-Op Week 3 and 4, the Impact Load demonstrated a declining trend at these time points. Then, the overall trend in both PROMs and IMU scores became similar again at Post-Op Week 5. Although this is just one example, it may represent a case in which traditional PROMs fail to identify a struggling patient during post-operative recovery and highlights the utility of RPM with IMUs to capture the post-operative progress of patients.

### 3.3. Pre-Op and Post-Op Week 6 6MWT and Maximum Knee Flexion Angle

The maximum distance walked in the 6 Minute Walk Test was 372.14 ± 61.57 m pre-operatively and 417 ± 136.32 m at Post-Op Week 6, but this did not reach significance (*p* = 0.688). The maximum knee flexion angle, measured with IMU sensors during walking and sit-to-stand exercises, went from 91.45 ± 1.401 degrees pre-operatively to 99.25 ± 9.59 degrees at Post-Op Week 6 but this not significantly different (*p* = 0.1563) (Figure 6).

## 4. Discussion

We present a scalable RPM workflow system to quantitatively assess post-operative function in patients’ recovery after knee arthroplasty. This novel platform combines traditional PROMs collection with wearable IMU sensors. It has been designed to easily incorporate into the routine clinical pathway in a low-maintenance and low-cost manner that is acceptable to patients. The simple use of an ankle bracelet system reliably captures data not previously available to the clinical team. Specifically, functional data from the community of degree of mobility (Bone Stimulus and Impact Load), impact load asymmetry and continuous range of motion are monitored and can be tracked over time. This represents a step change in monitoring post-operative knee replacement patients. Progressive charting of improved loading of the knee, symmetry of movement and range of movement allows for remote analysis of progress, compared to the traditional single visit or ‘6-week check-up appointment’. Furthermore, the use of IMUs abolishes both subjective and objective bias in measuring functional outcome in combination with well-established PROMs (OKS and EQ-5D/EQ-VAS) and PBOMs (6MWT). This enables a quantitative evaluation of patients’ recovery after surgery in physiotherapy clinics and the home environment. These pilot data establish a workflow system for future studies to measure post-operative recovery.

By providing accurate measurement of linear accelerations and angular velocities, single or multiple IMUs can be combined with biomechanical models to provide estimates of general activity [8], spatiotemporal gait metrics [9] and knee range of motion (ROM) [10]. Many PBOMs including sit-stand, stair climbing, or step-up tests, have also been accurately captured using IMUs [7,34]. In this study, in certain patients, there were notable discrepancies between subjective PROMs and objective assessments with IMU-based estimates of impact loading and Bone Stimulus. This highlights the need to include both metrics in a combined PROMs, PBOMs and wearable IMU metric. Furthermore, it might help explain the absence of differences in outcome when using the OKS in isolation [37,38,39]. Further validation is needed to assess the efficacy of IMU as an outcome measurement tool.

Monitoring the patient’s recovery after knee arthroplasty is important as improvements in actual physical activity have been shown to lead to reductions in perioperative adverse events, as well as joint pain and stiffness [40,41]. There is reported to be poor correlation between patient-reported and objectively assessed physical function [4,42,43,44]. PROMs have been shown to over-report the level of actual physical function and a ceiling effect has frequently been observed, whereby scores fail to discriminate patient outcomes [4,42,45]. The workflow presented in this study combines all three categories (PROMs, PBOMs and wearable IMUs) to capture the post-operative progress of patients.

The current system of capturing patient data relies on ‘face-to-face’ consultations or mailed questionnaires is an inefficient, time-consuming and costly process which captures only a sub-portion of the patient’s progress. Clinician follow-up is minimal (typically at 2 and 6 weeks post-surgery) and provides limited time to assess post-operative function. With the introduction of Enhanced Recovery After Surgery protocols, hospital length of stay has reduced, thereby decreasing the time for inpatient rehabilitation post-surgery [46]. After discharge, physiotherapy is commonly only provided weekly within a group setting after knee arthroplasty [47] and there is often poor continuity in the care teams’ documenting progress [48].

This RPM system allows surgeons and care teams to better understand their patients’ recovery with a wide range of subjective, objective, joint-specific and mobility data outputs. Individualized IMU sensor data from the home and physiotherapy settings are aggregated with PROMs and PBOMs on a weekly basis. The recent restrictions on patient contact related to the COVID-2019 pandemic has led to the rapid rise of telemedicine clinics and there is a strong need for more extensive adoption of RPM and wearable technologies [49,50]. This study supports the efficacy of RPM in orthopaedics.

This study has several limitations. Firstly, the data represent a small pilot cohort, so multivariate analysis was not possible to compare patient (i.e., age, BMI, co-morbidity) and surgical (i.e., TKA vs. UKA) factors. Secondly, the patients were given no instructions of what activities to perform at home and durations of activity varied, thus variation in amount and type of community-based activity captured with RPM may under- or over-estimate patients’ actual level of function. Thirdly, the IMU Step application captures patients’ walking behaviour only and does not account for stair climbing and other activities. Despite these limitations, this pilot study represents an initial validation of this wearable sensor technology.

## 5. Conclusions

We demonstrate a novel RPM system to provide a reliable and low-maintenance workflow to remotely monitor post-operative progress in knee arthroplasty patients. IMU outputs relating to mobility (Bone Stimulus and Impact Load), impact load asymmetry and range of motion can be obtained using commercially available IMU sensors. Through combining IMU data with established PROMs, a more accurate picture of the patient’s recovery after knee arthroplasty surgery can be established. Although these findings are promising, more research is required to directly correlate the IMU sensor outputs with patient outcomes.

## Figures and Tables

**Figure 1 sensors-21-05143-f001:**
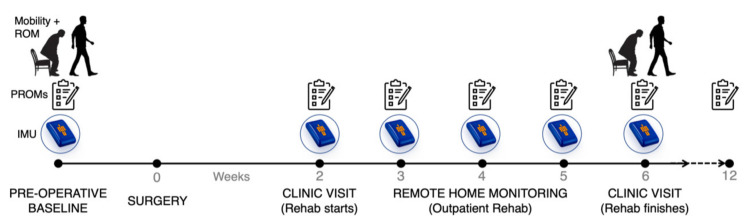
Schematic representing our workflow for post-operative remote patient monitoring of patients following total knee arthroplasty using inertial measurement units (IMUs). Mobility and knee flexion range of motion (ROM) were taken at baseline and 6 weeks post-op. Patient-reported outcome measures (PROMs) were taken at each time point as well as at 12-weeks post-op.

**Figure 2 sensors-21-05143-f002:**
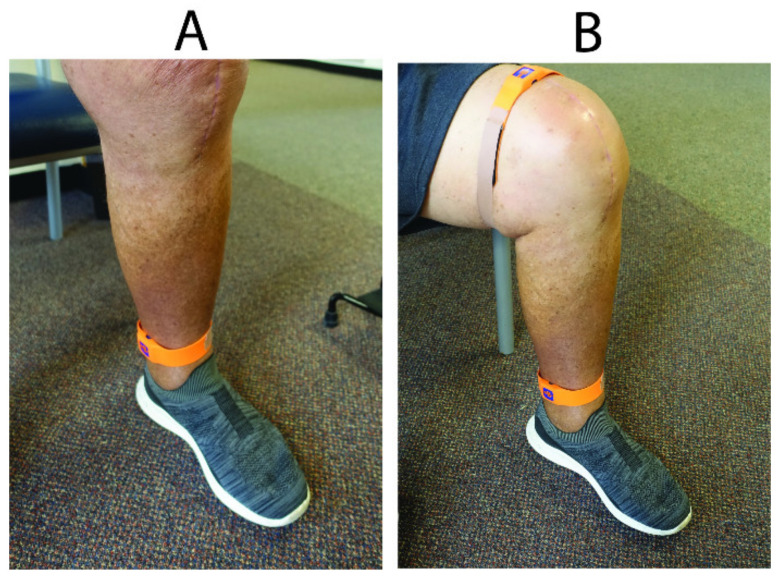
Inertial measurement unit (Vicon Blue Trident sensor) strapped to the ankle of the patient (**A**), as worn during weekly remote home monitoring after outpatient rehabilitation clinics. An additional thigh-worn sensor (**B**) was taped to the distal thigh pre-operatively and at Post-Op Week 6 to determine maximum knee flexion ankle.

**Figure 3 sensors-21-05143-f003:**
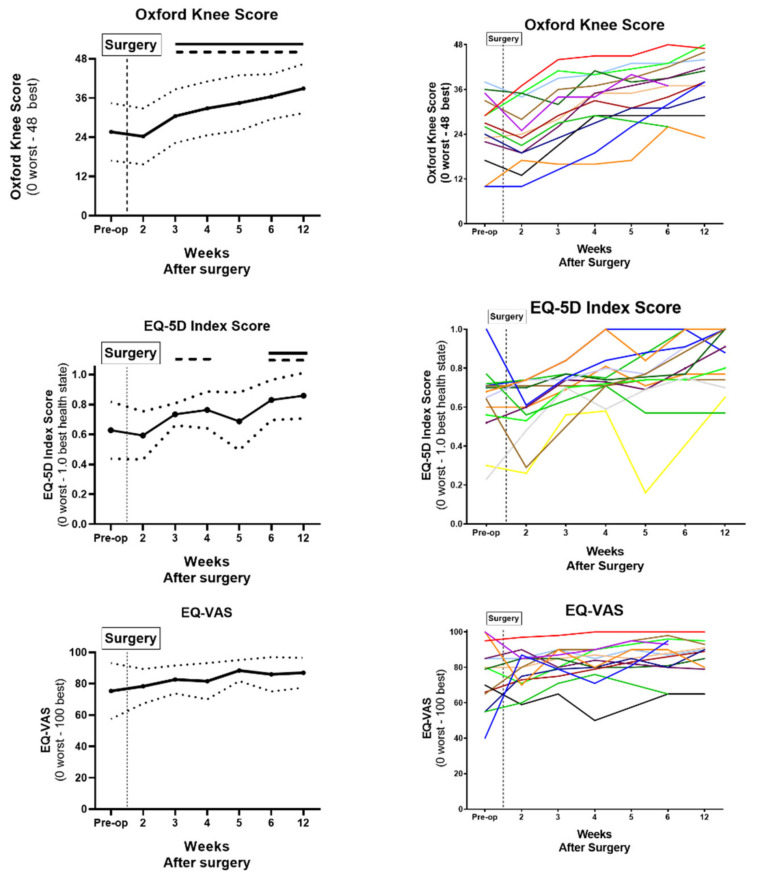
PROMs (OKS, EQ-5D Index Score and EQ-VAS) were recorded pre-operatively until Post-Op Week 12 presented as [left] mean (solid line) ± S.D (dotted line) and [right] all individual patient data represented by a different colour line. Data were analysed by one-way ANOVA with Tukey’s post-hoc analysis (*p*  <  0.05, comparison vs. Pre-Op Score [solid bar above] and *p* < 0.05, comparison vs. Post-op Week 2 Score [dashed bar above]).

**Figure 4 sensors-21-05143-f004:**
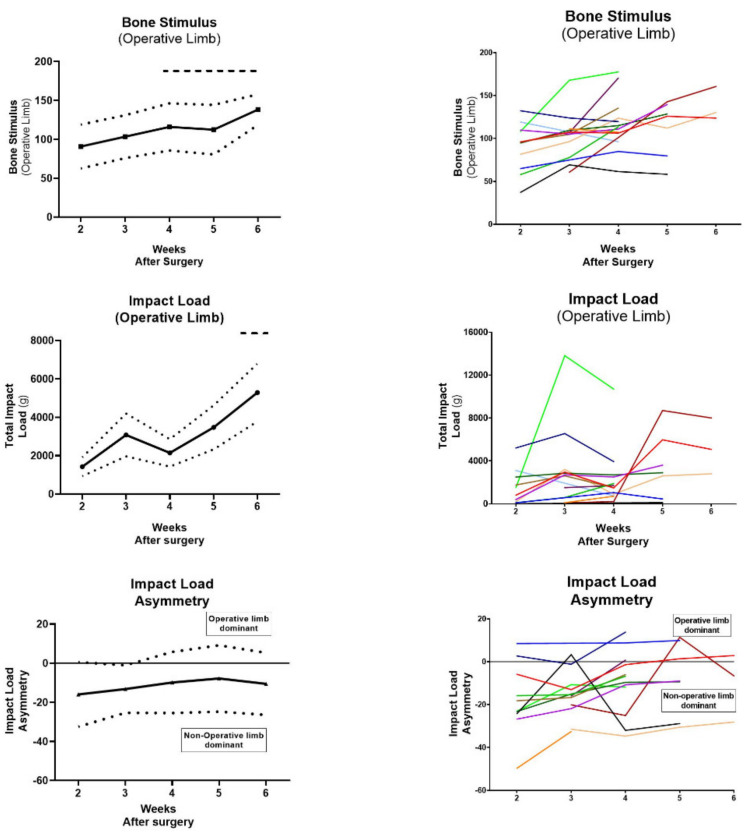
IMU Outputs (Bone Stimulus, Impact Load and Impact Load Asymmetry) presented as [left] mean (solid line) ± S.D (dotted line) and [right] all individual patient data represented by a different colour line. Data were analysed by one-way ANOVA with Tukey’s post-hoc analysis (*p* < 0.05, comparison vs. Post-op Week 2 Score [dashed bar above]).

**Figure 5 sensors-21-05143-f005:**
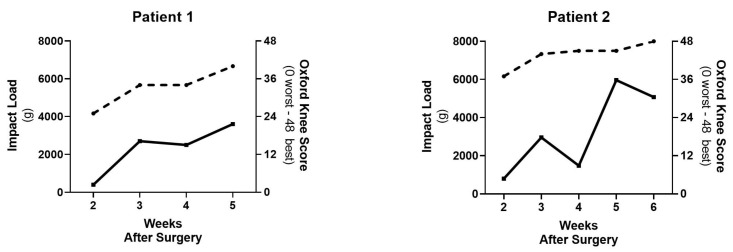
Individual data for Patient 1 and 2 with PROMs scores (OKS, dashed line) and IMU Outputs (Impact Load, solid line) over the post-operative period.

**Figure 6 sensors-21-05143-f006:**
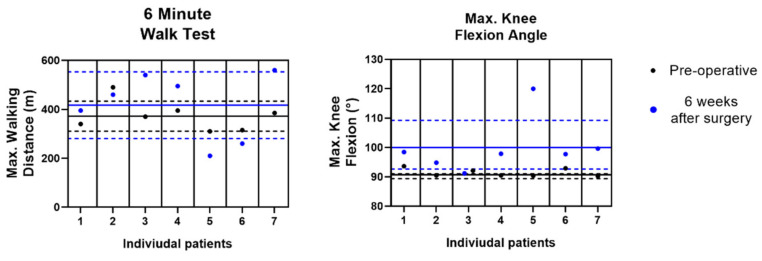
The distance walked over the 6 Minute Walk Test and Maximum Knee Flexion Angle (measured with IMU sensors) pre-operatively and at Post-Op Week 6. The mean (solid line) ± S.D. (dotted line) all individual patient data are presented.

**Table 1 sensors-21-05143-t001:** Patient Demographics and Operation Details. Data presented as mean ± SD.

Demographics and Operation Details	Patient Details
Gender	6 Males, 8 Females
Age	66.8 ± 7.0 years
BMI	30.6 ± 5.9 kg/m^2^
Arthroplasty Type	9 TKA, 5 UKA
ASA Classification	2.1 ± 0.7

## Data Availability

The data presented in this study are available on request from the corresponding author.

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
