# Peer review of "Remote Patient Monitoring with Wearable Sensors Following Knee Arthroplasty"

_sensors, 2021, doi:10.3390/s21155143_

Round 1

Reviewer 1 Report

The manuscript reports a novel way of validating knee functional assessment following knee arthroplasty using the means of inertial measurement unit. This is cross-referenced with traditional assessment using a series of questionnaires and gait measurements. In my opinion, the report demonstrated the capability of digital measurements as means of objective functional testing. Overall, I think the manuscript serves its merits. To further improve the manuscript, I have the following comments:

1) The difference between pre-op and post-op 6 minute walk test and maximum flexion angle is not significant possibly due to the small number of participants. Rather than reporting statistical difference on a small population, it might be worthwhile to note individual walking distance and knee flexion angle. 

2) It would be worthwhile to consider using traditional goniometer to compare maximum flexion angle measured using the IMU.

3) There are some typos in the manuscript, for example in line 164 compliant, instead of complaint. 

Author Response

Thank you for reviewing the manuscript and your comments.

1) We appreciate the reviewers concerns regarding the small number of participants in the study (particularly for the maximum knee flexion angle and 6 minute walk test) and the limitations associated with doing statistical analysis in a small cohort. In response to the suggestion of reporting on individual differences we have generated some figures to show individual responses in Figure 6.

2)  Although we appreciate the ease at which a goniometer can be used to compare maximum knee flexion angle with IMUs, the intent here is to measure functional, dynamic angles (during walking and sit-to-stand exercises) which is only possible with the IMU. Of course the IMU can be used to replace a traditional goniometer, but there are also errors associated with goniometer measurements that can be improved with an appropriate callibration with IMUs.

3) Our sincere apologies for the typing mistakes in the manuscript. We have reviewed the entire manuscript again and corrected any errors.

Reviewer 2 Report

A Novel RPM of IMU is proposed to be used in conjunction with PROM and PBOM. This is an interesting validation of the use of IMU's parameters obtained remotely through wearable sensors. However I don't know if  it will add any more clinically relevant insights in assessing arthroplasty outcome that standard POM and PBOM standards already provide. Also ease of use might be different as well as recall bias might be confounding factors as patients might remove them intermittently for comfort. Nonetheless I recommend publishing this work after some minor spelling checks.

Author Response

Thank you for reviewing the manuscript and your comments. 

We acknowledge that this is a pilot study of IMU wearable sensor technology. More research is required to directly correlate the IMU sensor outputs directly with patient outcomes after knee arthroplasty. This is stated in our limitations section and stated in line 333-334 of our manuscript. 

We apologise for the typing errors and have reviewed the manuscript closely again.

Reviewer 3 Report

General considerations

This study presents results of improvement of walking abilities of patients undergoing total knee arthroplasty (TKA) or unicompartmental knee arthroplasty (UKA) surgery, measured with traditional self-reported evaluation scales and wearable sensors (IMUs). The manuscript is well written and conveys a clear message. It reports application of inertial sensors for the evaluation of physiological quantities. However, the way the study is presented in the current version of the manuscript does not make it suitable for Sensors journal. The study focuses on physiological effects and it is very brief and rushed on the sensor signal analysis, which is the more interesting content for Sensor journal readers. In other words, although improvement related to TKA and UKA are a finding of sure interest in journals of the field, the manuscript does not show a tangible scientific improvement of the current knowledge in the inertial sensor field. From the reading, I understood that authors dealt extensively with data analysis for obtaining their final results, in a way that, although not so original, maybe of interest if extensively explained.

My suggestion is to amend the manuscript covering the major issues below reported, and resubmit a version which definitely would be more suited for the journal.

Major issues:

  1. It is not clear how the quantity “total impact load” was measured. As it’s expressed in [g] one might think that it is the maximum value of the signal during the impact. Please, clarify better. Was it the acceleration in a specific direction? The module of the three axis? Was it normalized with respect to the bodymass of the subject which might have changed after surgery?
  2. Bone Stimulus is even more unclear. I see references to other studies, but I strongly suggest authors to explain their analysis methodology to be suited for this journal.
  3. Authors report that a machine learning algorithm was used for predicting maximum knee flexion angle. Predicting from what? Why? What is the performance of the machine-learning alghoritm? How accurate was it in this specific study? Was the accuracy acceptable? How was it assessed?

Author Response

Thank you for reviewing the manuscript and your thoughtful comments. 

  1. We now state the formula used to calculate total impact load in lines 139-143 to clarify this. We performed pre-operative anlaysis within 1-2 weeks before surgery, so we do not expect the body mass of the subject to change significantly over the 8 week (2 week pre-operative and 6 week post-operative) measurement period to warrant normalisation.
  2. We now state the formula used to calculate Bone Stimulus in the manuscript and give a more exact description/definition of the metric in lines 144 to 151.
  3. We now give a more complete description of how the machine learning algorithm was developed and implemented in lines 171 - 183. A previous study has already described this methodology and is referenced in the manuscript "Brouwer NP, Yeung T, Bobbert MF, Besier TF (2021) 3D trunk orientation measured using inertial measurement units during anatomical and dynamic sports motions. Scand J Med Sci Sport 31:358–370 https://doi.org/10.1111/sms.13851

Round 2

Reviewer 3 Report

Authors covered the issues in a almost satisfactory manner. My only last concern is still about the machine learning algorithm. I still think they should clarify better which data they used for training. Knee angles? How measured? This passage is still unclear.

Author Response

Thank you for your additional comments and thorough feedback. 

We have now briefly stated how the machine learning algorithm was trained to determine knee flexion angle in lines 174 to 183 of the manuscript.

Please note that the complete methods for the training process for this surrogate model are available in a conference proceeding, which is now referenced in the manuscript.